# Tissue-Specific RNA-Seq Analysis of Cotton Roots’ Response to Compound Saline-Alkali Stress and the Functional Validation of the Key Gene *GhERF2*

**DOI:** 10.3390/plants14050756

**Published:** 2025-03-01

**Authors:** Aiming Zhang, Qiankun Liu, Xue Du, Baoguang Xing, Shaoliang Zhang, Yanfang Li, Liuan Hao, Yangyang Wei, Yuling Liu, Pengtao Li, Shoulin Hu, Renhai Peng

**Affiliations:** 1College of Agricultural, Tarim University, Alar 843300, China; m904780569y@163.com (A.Z.); 19937820150@163.com (B.X.); zsl001224@icloud.com (S.Z.); sybks10lyf25@163.com (Y.L.); hla19971016@163.com (L.H.); 2School of Biotechnology and Food Engineering, Anyang Institute of Technology, Anyang 455000, China; duxue101199@163.com (X.D.); daweianyang@163.com (Y.W.); liuylay2012@163.com (Y.L.); 3The Ministry of Agriculture, Institute of Cotton Research, Chinese Academy of Agricultural Sciences, Anyang 455000, China; liuthundering@163.com

**Keywords:** RNA-seq, cotton root, saline-alkali stress, plant hormone signal transduction, MAPK signaling, *GhERF2*

## Abstract

Saline-alkali stress is one of the major abiotic stresses threatening crop growth. Cotton, as a “pioneer crop” that can grow in saline and alkali lands, is of great significance for understanding the regulatory mechanisms of plant response to stresses. Upland cotton has thus become a model plant for researchers to explore plant responses to saline-alkali stresses. In this study, RNA sequencing was employed to analyze tissue-specific expression of root tissues of TM-1 seedlings 20 min after exposure to compound saline-alkali stress. The RNA-Seq results revealed significant molecular differences in the responses of different root regions to the stress treatment. A total of 3939 differentially expressed genes (DEGs) were identified from pairwise comparisons between the non-root tip and root tip samples, which were primarily enriched in pathways including plant hormone signal transduction, MAPK signaling, and cysteine and methionine metabolism. Combined with the expression pattern investigation by quantitative real-time PCR (qRT-PCR) experiments, a key gene, *GhERF2* (*GH_A08G1918*, ethylene-responsive transcription factor 2-like), was identified to be associated with saline-alkali tolerance. Through virus-induced gene silencing (VIGS), the *GhERF2*-silenced plants exhibited a more severe wilting phenotype under combined salt-alkali stress, along with a significant reduction in leaf chlorophyll content and fresh weights of plants and roots. Additionally, these plants showed greater cellular damage and a lower ability to scavenge reactive oxygen species (ROS) when exposed to the stress. These findings suggest that the *GhERF2* gene may play a positive regulatory role in cotton responses to salt-alkali stress. These findings not only enhance our understanding of the molecular mechanisms underlying cotton response to compound saline-alkali stress, but also provide a foundation for future molecular breeding efforts aimed at improving cotton saline-alkali tolerance.

## 1. Introduction

Soil salinization is a significant environmental challenge facing humanity, posing a growing threat to the sustainable development of agricultural ecosystems worldwide [1]. Currently, approximately 20% of the arable land around the world is affected by adverse environments, with 954 million hectares affected by salinity, and this trend is increasing annually [2,3]. Soil salinization could lead to land degradation, severely reducing agricultural productivity, imposing serious constraints on sustainable agricultural development, potentially triggering food supply security issues [4]. Therefore, how to develop tolerant varieties to potentially utilize saline soils is one of the major challenges for all breeding scientists. Long-term agricultural practices have demonstrated that they are among the most effective methods to develop salt-tolerant crops for reclaiming saline soils [5]. This approach not only directly addresses the soil salinity problems, but also provides a strategy to enhance agricultural resilience [6] for sustainable agricultural development.

Xinjiang is a typical inland arid region in China characterized by low rainfall and high evaporation, resulting in high soil salinity. The salts there are mainly composed of cations (including Na^+^, K^+^, Ca^2+^, and Mg^2+^) and anions (including Cl^−^, SO_4_^2−^, HCO^3−^, and CO_3_^2−^) [7]. Previous research has indicated that soil salinization is divided into two distinct types: neutral salt stress, primarily incurred by NaCl and Na_2_SO_4_, is referred to as salt stress [8], and alkaline salt stress, incurred by Na_2_CO_3_ and NaHCO_3_, as alkali stress [9,10]. Although closely related, salt and alkali stresses are distinct in nature. Compared to salt stress, alkali stress not only causes severe ionic toxicity and osmotic stress in plants [11,12] but also induces high pH stress [13]; hence, it imposes a greater impact on ecological damage. Besides, oxidative stress resulting from osmotic stress and ionic toxicity may also damage vital biological components, such as membrane lipids, proteins, and nucleic acids in plant cells [14,15]. It can even damage the integrity of cellular structures, ultimately leading to physiological and metabolic disorders and significantly inhibiting plant growth and development [16,17,18].

Plant roots provide anchorage, absorption, storage, and transport of minerals and water [19]. They facilitate communication and interaction between plants and soil microbiomes as well as between plants [20]. Roots exhibit high developmental plasticity and can adapt to different environmental conditions [3,21]. Different root regions play distinct roles in plant growth and development, with root tips as the young end sections [22]. When exposed to different environmental factors, such as water, salinity stresses, pH toxicity, and mechanical impedance, the root tips often undergo the earliest physiological and biochemical changes, triggering corresponding molecular regulatory mechanisms [23,24].

Cotton (*Gossypium* spp.) has long been recognized as a salt-tolerant crop and is often planted as a “pioneer crop” in saline soils. Cotton cultivation is very important for the development and utilization of saline lands and the advancement of the cotton industry. Cotton fiber is an important raw material for the global textile industry, accounting for 35% of total global fiber consumption [25,26]. Currently, cotton is grown in over 80 countries, with cotton being the major commercial crop in more than 30 of them, exerting a significant impact on the global economy annually [27,28]. Although its inherent salt tolerance has made positive contributions to the improvement of saline soil, cotton still faces challenges, including low yield, high production costs, and low profitability under saline soil conditions [29,30].

Transcriptome sequencing (RNA-seq) aids in identifying gene expression profiles, differentially expressed genes (DEGs), and their transcriptional regulatory patterns. By analyzing multiple samples across different developmental stages, tissues, or adverse conditions, RNA-seq rapidly provides gene expression information and identifies new genes mediating responses to various physiological processes in plants, elucidating the underlying molecular mechanisms of these processes [31]. These studies have successfully been applied to crops like cotton [32], jujube [33], and sorghum [34], highlighting their potential role in regulating plant responses to saline-alkali stress [28]. Notably, in a study focusing on cotton roots with different potassium ion affinities, a key regulatory target for K^+^ uptake under potassium deficiency was successfully identified using RNA-seq technology [35]. Previous studies also identified a key gene, *GhMKK3*, in cotton roots through RNA-seq and quantitative real-time PCR (qRT-PCR). This gene plays a role in the drought stress response by regulating cotton stomatal behavior and root hair growth [36]. When cotton is exposed to complex and varied secondary saline-alkali environments in open fields, it undergoes various physiological and biochemical reactions, as well as changes in related metabolic pathways [37]. However, the responses of different root regions of cotton to simulated secondary saline-alkali stress have not been reported.

In this study, a secondary saline-alkali stress environment was simulated indoors using specific compound ratios. Samples from different root regions were collected and subjected to RNA-seq with three biological replicates. By pairwise comparing these samples, a large number of DEGs were identified. These DEGs were further subjected to functional enrichment analyses of Gene Ontology (GO) and the Kyoto Encyclopedia of Genes and Genomes (KEGG). Comprehensive analysis indicated that plant hormone signal transduction, MAPK signaling pathways, and cysteine and methionine metabolism might make significant contributions to cotton resistance against saline-alkali stress [38,39]. Using qRT-PCR, a key gene, *GH_A08G1918* (ethylene-responsive transcription factor 2-like, *GhERF2*), was identified to play a role in cotton tolerance to saline-alkali stress. Previous studies have shown that ERFs from different plant species play significant roles in regulating plant growth, development, and immunity, and responses to cope with various abiotic stresses in complex environments [40,41,42]. Virus-induced gene silencing (VIGS) of *GhERF2* revealed that the silenced cotton plants exhibited increased sensitivity to saline-alkali stress. These findings enhance our understanding of the transcriptional regulation of saline-alkali resistance genes in cotton, providing a basis for molecular breeding and molecular mechanism research on cotton saline-alkali tolerance. This work also facilitates future progress in precision gene editing to aid in cotton breeding.

## 2. Results

### 2.1. Transcriptome Sequencing and Data Quality Assessment

To systematically understand the key signaling pathways in the roots of upland cotton TM-1 under compound saline-alkali stress and to identify critical genes responding to this stress, 12 cDNA libraries were constructed and subjected to high-throughput sequencing, including the root tip sections (RTS-1, RTS-2, RTS-3) and non-root tip sections (NRTS-1, NRTS-2, NRTS-3) of cotton seedlings subjected to 20 min of compound saline-alkali stress, and control seedlings treated with water for 20 min were sampled in the same manner for root tip (RTCK-1, RTCK-2, RTCK-3) and non-root tip (NRTCK-1, NRTCK-2, NRTCK-3).

In total, approximately 288.451 million clean reads were obtained in this study, with an average of about 24.1 million clean reads per sample. The GC content for each sample was around 45%, and the Q20 values ranged from 96.01% to 96.72% (Table 1). These results directly illustrated that the sequencing data are authentic and reliable for further analysis.

Based on the annotated reference genome, a total of 72,761 expressed genes were identified in the RNA-Seq dataset, of which the expression levels were subsequently assessed using fragments per kilobase of transcript per million mapped reads (FPKM). Principal component analysis (PCA) on all 12 samples (Appendix A) showed that the first and second principal components explained 38.4% and 22.5% of the variance, respectively, indicating that the replicates of the control group (NRTCK and RTCK) and the treatment group (NRTS and RTS) were well clustered, and that samples from different groups, whether control or treatment, showed clear separation. Additionally, Pearson correlation coefficient (PCC) analysis of the entire sample set revealed that the gene expression similarity among the three biological replicates within each treatment group exceeded 83% (Appendix A). All these analyses demonstrated that the samples are of high quality and reproducibility, making them suitable for subsequent differential expression and functional enrichment analyses.

### 2.2. Identification and Functional Enrichment Analysis of DEGs

To uncover the differences in gene expression between the control and treatment groups, DEGs were screened by comparing the expression profiles of samples from the same sampling root sites (RCK-RS and RTCK-RTS). Specifically, compared with the corresponding root sites of the control group, the non-root tip sites of treatment had 5702 up-regulated and 4101 down-regulated DEGs (Figure 1A), which were enriched in three GO categories (Figure 1B), namely biological process (BP), molecular function (MF), and cellular component (CC). In the BP category, both the up-regulated and down-regulated DEGs were enriched in the same top three terms, namely metabolic process (MP) (GO:0043170), cellular process (GO:0009987), and single-organism process (GO:0044699). In the MP term, catalytic activity (GO:0003824) enriched the largest number of up-regulated DEGs, while binding (GO:0005488) was more correlated with down-regulated DEGs. In the CC category, the top enriched terms were membrane (GO:0016020) and cell (GO:0005623), as well as their parts (GO:0044425 and GO:0044464). KEGG pathway annotation of these DEGs revealed that most of the up-regulated DEGs were significantly clustered in the metabolic pathway (ko01100), ascorbate and aldarate metabolism (ko00053), plant–pathogen interaction (ko04626), amino sugar and nucleotide sugar metabolism (ko00520), and carbon fixation in photosynthetic organisms (ko00710) (Figure 1C). The down-regulated DEGs were clustered in the top five KEGG pathways: circadian rhythm–plant (ko04712), basal transcription factors (ko03022), diterpenoid biosynthesis (ko00904), plant hormone signal transduction (ko04075), and cysteine and methionine metabolism (ko00270) (Figure 1D).

In the root tip comparison, 3717 up-regulated and 3856 down-regulated DEGs were identified in the treatment samples compared to the control (Figure 2A), and the enriched GO terms were similar to those in non-root tip samples (Figure 2B). For example, the terms enriched with the highest number of DEGs were MP (GO:0043170), cellular process (GO:0009987), and single-organism process (GO:0044699) in the BP category; catalytic activity (GO:0003824) and binding (GO:0005488) in the MF category; and membrane (GO:0016020) and cell (GO:0005623) in the CC category. KEGG annotation of these DEGs revealed that the up-regulated DEGs were significantly clustered into the pathways of ascorbate and aldarate metabolism (ko00053), plant–pathogen interaction (ko04626), lysine biosynthesis (ko00300), metabolic pathways (ko01100), and biosynthesis of secondary metabolites (ko01110) in the former (Figure 2C); and that the down-regulated DEGs were clustered into plant hormone signal transduction (ko04075), spliceosome (ko03040), phosphatidylinositol signaling system (ko04070), inositol phosphate metabolism (ko00562), and valine, leucine, and isoleucine biosynthesis (ko00290) (Figure 2D).

### 2.3. Key DEG Screening and Functional Enrichment Investigation

To screen key DEGs related to saline-alkali resistance, a Venn diagram was created to analyze the overlap of DEGs in different segments of seedling roots after compound saline-alkali stress treatment. The results showed that 3634 DEGs were commonly identified among all four samples, while 6169 and 3939 ones were identified exclusively in the non-root tip and root tip regions, respectively (Figure 3A). GO enrichment analysis (Figure 3B) revealed that most of the common DEGs were enriched in hydrogen peroxide catabolic process (GO:0042744), regulation of transcription, DNA-templated (GO:0006355), and regulation of nucleic acid-templated transcription (GO:1903506) in the BP category; in intrinsic component of membrane (GO:0031224), integral component of membrane (GO:0016021), and membrane part (GO:0044425) in the CC category; and DNA binding (GO:0003677) and oxidoreductase activity, acting on peroxide as acceptor (GO:0016684) in the MF category. KEGG pathway analysis showed that they were significantly clustered into five pathways, namely metabolism, genetic information processing, environmental information processing, cellular processes, and organismal systems (Figure 3C). In the metabolism pathways, DEGs were primarily enriched in global and overview maps, carbohydrate metabolism, and lipid metabolism. In the pathways of transcription, signal transduction, transport and catabolism, and environmental adaptation, DEGs were primarily enriched in genetic information processing, environmental information processing, cellular processes, and organismal systems, respectively.

Functional enrichment of the tissue-specific DEGs revealed that DEGs in the non-root tip samples were enriched in the top three GO terms of cell wall biogenesis (GO:0042546), cell wall organization or biogenesis (GO:0071554), and polysaccharide metabolic process (GO:0046906) in the BP category; in intrinsic component of membrane (GO:0031224), membrane part (GO:0044425), and integral component of membrane (GO:0016021) in the CC category; and in tetrapyrrole binding (GO:0046906), UDP-glucose 6-dehydrogenase activity (GO:0003979), and heme binding (GO:0020037) in the MF category (Appendix A). The top five KEGG pathways in which these DEGs clustered included amino sugar and nucleotide sugar metabolism (ko00520), metabolic pathways (ko01100), biosynthesis of secondary metabolites (ko01110), beta-alanine metabolism (ko00410), and carbon metabolism (ko01200) (Appendix A). The DEGs in the root tip samples were significantly enriched in the GO terms of positive regulation of transcription, DNA-templated (GO:0045893), positive regulation of RNA metabolic process (GO:0051254), and positive regulation of RNA biosynthetic process (GO:1902680) in the BP category; in nucleic acid binding (GO:0003676), ER retention sequence binding (GO:0046923), and 1,3-beta-D-glucan synthase activity (GO:0003843) in the MF category; and in 1,3-beta-D-glucan synthase complex (GO:0000148) and chromatin (GO:0000785) in the CC category (Appendix A). KEGG pathway analysis indicated that the root tip DEGs were significantly clustered in the pathways of plant hormone signal transduction (ko04075), lysine biosynthesis (ko00300), ribosome biogenesis in eukaryotes (ko03008), monobactam biosynthesis (ko00261), MAPK signaling pathway–plant (ko04016), and cysteine and methionine metabolism (ko00270) (Appendix A), implying that these pathways might participate in the cotton response to compound saline-alkali stress.

### 2.4. Analysis of Key Pathways Related to Compound Salinze-Alkali Stress

Through functional enrichment analysis of DEGs specifically expressed in root tips, it was preliminarily suggested that plant hormone signal transduction, MAPK signaling pathway–plant, and cysteine and methionine metabolism were associated with compound saline-alkali stress. A total of seventy-four DEGs were identified within the eight key pathways of plant hormone signal transduction (Figure 4 and Appendix A), including five DEGs related to abscisic acid (all down-regulated)— these DEGs belong to protein phosphatase 2C (PP2C) and serine/threonine-protein kinase SRK2 (SNRK2); nine DEGs to ethylene (two up-regulated and seven down-regulated ones)—these DEGs belong to serine/threonine-protein kinase (CTR1), ethylene-insensitive protein 3 (EIN3), and ethylene-responsive transcription factor 1 (ERF1/2); twenty-three DEGs to auxin (six up-regulated and seventeen down-regulated ones)—these DEGs belong to auxin influx carrier (AUX1), auxin-responsive protein IAA (IAA), and SAUR family protein (SAUR); nine DEGs to brassinosteroid (five up-regulated and four down-regulated ones)—these DEGs belong to brassinosteroid insensitive 1-associated receptor kinase 1 (BAK1), BRI1 kinase inhibitor 1 (BKI1), and brassinosteroid resistant 1/2 (BZR1/2); twenty-two DEGs to gibberellin (all the down-regulated)—these DEGs belong to DELLA protein (DELLA) and phytochrome-interacting factor 4 (PIF4). These findings imply that root tips respond to saline-alkali stress through multiple plant hormone pathways.

In the MAPK signaling pathway, a total of 15 down-regulated DEGs were identified in the H_2_O_2_ pathway (Figure 5)—these DEGs belong to LRR receptor-like serine/threonine-protein kinase FLS2 (FLS2), WRKY transcription factor 33 (WRKY33), WRKY transcription factor 22 (WRKY22), senescence-induced receptor-like serine (FRK1), mitogen-activated protein kinase 1/2 (MPK1/2), and pathogenesis-related protein 1 (PR1). Interestingly, four key genes were commonly identified in both the plant hormone signal transduction and MAPK signaling pathways (*GH_D05G0019*, *GH_D08G0941*, *GH_A02G0431*, and *GH_D10G1752*), all of which were found to participate in the ethylene pathway. Hence, it was speculated that the ethylene pathway might play a crucial role in the plant’s response to compound saline-alkali stress. In addition, twelve DEGs were identified in the cysteine and methionine metabolism pathway (Figure 5), of which five were up-regulated and seven down-regulated. The variations in the expression levels of these key genes may have an impact on the sulfur metabolism and biosynthesis of terpenoid backbone compounds in plants.

### 2.5. Validation of RNA-Seq Data Using qRT-PCR Experiment

To verify the validity of the RNA-seq data, a qRT-PCR experiment was conducted on twelve randomly selected DEGs. The qRT-PCR results showed that the relative expression profiles of these genes were strongly positively correlated with their FPKM values (Figure 6). The consistent expression trends confirmed that the RNA-seq data were reliable and valuable for further functional verification of candidate genes related to plant resistance.

### 2.6. Expression Pattern Analysis of GhERF2 Under Compound Saline-Alkali Stress

Transcription factors (TFs) play a crucial role in transcriptional regulation processes within the cell nucleus, and nuclear localization is one of the key characteristics for determining whether a gene functions as a TF or not. In plants, there are multiple TF families, each playing distinct roles in stress responses. A TF gene commonly enriched in both ethylene signal transduction and MAPK signaling pathway, *GhERF2* (*GH_A08G1918*), showed diverse expression variations in both the non-root tip and root tip samples under saline-alkali stress; it manifests as a decrease in the expression level of the target gene after 20 min of salt-alkali stress, implying its potentially important role in plant resistance. To investigate whether this TF gene plays its role in transcriptional regulation within the plant cell nucleus, the coding sequence (CDS) of *GhERF2* was cloned into the *pCMBIA2300-DsRED2* vector, and a *35S::GhERF2-RFP* construct was successfully made. Subsequently, *Agrobacterium* carrying this recombinant vector was transiently transformed into tobacco leaves via leaf surface injection. After 48 h of dark incubation, observations using a laser confocal microscope showed that *35S::GhERF2-RFP* was specifically expressed in the nucleus, whereas the *35S::RFP* empty vector control was expressed in both the nucleus and cytoplasm (Figure 7A). This indicated that the target protein lacked a transmembrane transport structure, confirming that *GhERF2* exhibited distinct characteristics of a TF.

Since *GhERF2* was identified from the root tips of cotton seedlings under saline-alkali stress, its tissue-specific expression in roots, stems, flowers, and leaves of TM-1 seedlings was examined (Figure 7B). The results showed that *GhERF2* was highly expressed in the stem and ovule of cotton, with a considerable expression in the roots.

Additionally, qRT-PCR experiments were also utilized to examine the responding changes of *GhERF2* in cotton root at various time points (0 h, 1 h, 3 h, 6 h, 12 h, 24 h, 48 h, 72 h) after saline-alkali stress treatment (Figure 7C). It was observed that saline-alkali stress strongly induced the expression of *GhERF2* from 0 h to 24 h, reaching its peak at 24 h. Then its expression in cotton leaves gradually declined till 72 h. The results indicated a significant regulation of *GhERF2* expression in response to these stresses.

### 2.7. Silencing of GhERF2 Reduces Cotton Tolerance to Saline-Alkali Stress

To further investigate the physiological role of *GhERF2* in cotton under saline-alkali stress, VIGS was employed to suppress *GhERF2* expression. Two weeks after *Agrobacterium* infection, *VIGS*-*GhCLA1* plants exhibited a noticeable albino phenotype, indicating the effectiveness of *VIGS* (Figure 8A). Subsequent qRT-PCR experiments showed that the expression of *GhERF2* in the roots of *TRV*:*GhERF2* plants was reduced by 60% compared to *TRV*:*00* plants, confirming the effective silencing of *GhERF2* (Figure 8B). After successful silencing, plants with consistent growth status were selected for compound saline-alkali stress treatment for further experiments.

After 48 h of compound saline-alkali stress treatment, leaves of WT, *TRV*:*GhERF2*, and *TRV*:*00* seedlings exhibited varying degrees of wilting. Compared with other treatments, *TRV*:*GhERF2* seedlings exhibited more significant and severe wilting (Figure 8C–H). The wilting degree of WT and *TRV*:*00* was similar. Then, biomass indicators of all treatments, including chlorophyll content, fresh weight, and dry weight, were measured. The results showed that under saline-alkali stress, there was a significant difference in the whole plant fresh weight between *TRV*:*GhERF2* and *TRV*:*00* plants, with *TRV*:*GhERF2* plants having significantly lower fresh weight after 48 h of stress (Figure 9A). However, plant height, root length, root fresh weight, plant dry weight, and root dry weight did not show significant differences between *TRV*:*GhERF2* and *TRV*:*00* plants (Appendix A). Chlorophyll content tests revealed that *TRV*:*GhERF2* plants had significantly lower chlorophyll content than TRV:00 plants after saline-alkali stress, while there was no substantial difference between *TRV*:*00* and WT plants (Figure 9B). When WT, *TRV*:00, and *TRV*:*GhERF2* plants were treated with a mixed solution of 200 mM sodium chloride and 150 mM NaHCO_3_ and Na_2_CO_3_ (molar ratio NaHCO_3_:Na_2_CO_3_ = 2:1), it was observed that *TRV:GhERF2* plants exhibited more severe wilting than *TRV*:*00* plants, which was similar to those observed under saline-alkali stress (Appendix A). Additionally, both chlorophyll content and whole plant fresh weight were lower to varying degrees in *TRV*:*GhERF2* plants than in *TRV*:*00* plants (Appendix A).

### 2.8. Physiological Characteristics of GhERF2-Silenced Cotton Plants

Salt stress often leads to an oxidative burst, causing damage to plants. Physiological indicators closely related to reactive oxygen species (ROS), including superoxide dismutase (SOD) activity and malondialdehyde (MDA) content, were evaluated. The results showed that under saline-alkali stress, SOD activity in *TRV*:*GhERF2* plants was significantly higher than in *TRV*:*00* plants (Figure 9C). In contrast, MDA content was significantly increased in *TRV*:*GhERF2* plants compared to *TRV*:*00* plants (Figure 9D). These findings further demonstrate that silencing *GhERF2* might reduce the ability of cotton seedlings to scavenge ROS, exacerbating the oxidative burst under stress conditions and making cotton more sensitive to saline-alkali stress.

Leaf staining with 3,3′-diaminobenzidine (DAB), which was also used to detect hydrogen peroxide levels, revealed that hydrogen peroxide levels in *TRV*:*GhERF2* plant leaves were significantly higher than those in *TRV*:*00* plant leaves (Figure 10). These results suggested that the ability of *GhERF2*-silenced plants to scavenge ROS was reduced.

## 3. Discussion

The increasing severity of soil salinization and alkalization is threatening the agricultural production and economy. With the continuous expansion of saline-alkaline soil, one effective measure for cotton production is to develop saline-alkali-tolerant varieties to promote the reuse and sustainable development of these lands [3,5]. However, the molecular mechanisms of cotton plant response to compound saline-alkali stress are not yet fully understood, despite the relatively abundant reports on this topic. The main objective of this study is to elucidate the molecular mechanism of cotton response to saline-alkali stress, and to identify key genes and regulatory pathways that make significant contributions to stress response. To achieve this, a secondary saline-alkaline field environment was simulated in the laboratory using a compound mixture and transcriptome sequencing was conducted on the roots of cotton seedlings after being treated with the mixture. The results supported that plant hormone signal transduction, MAPK signaling pathways, and cysteine and methionine metabolism play significant roles in cotton’s defense against compound saline-alkali stress. Understanding these pathways and identification of the key genes in the cascades of pathways will provide deeper insights into the molecular biological processes of cotton’s response to saline-alkali stress.

Transcriptome sequencing is frequently adopted to identify transcripts and new genes. This method has been widely applied in studies of different cotton varieties’ responses to abiotic stress [43,44,45,46]. In this study, via analysis of transcriptome data, a total of 136 DEGs were identified to be associated with the above-mentioned pathways. Given that cotton undergoes a series of complex regulatory processes in response to various stresses, these DEGs may participate in regulating plant response to compound saline-alkali stress.

### 3.1. Transcriptome Sequencing and Analysis

Auxin positively participates in controlling plant growth and development under various environmental conditions [47]. Studies have shown that changes in environmental conditions can alter the expression levels of auxin signaling pathway-related genes, such as *IAA22*, *IAA13*, *IAA27*, *IAA14*, and *IAA7*, thereby promoting root expansion in carrots [48]. In our study, several DEGs were identified relevant to the auxin pathway. These results suggested that the signaling of cotton auxin might undergo changes under environmental stress, thereby regulating root growth and development.

Gibberellin is considered a molecular clock for plant root development and makes a crucial contribution to regulating the development of root apical meristems [49,50]. Under compound saline-alkali stress, the gibberellin signaling pathway was inhibited, with 22 DEGs identified in this pathway showing down-regulated expression. Sixteen of these DEGs were related to DELLA proteins, significantly impacting the physiological activity of cotton roots. DELLA proteins are inhibitors in the gibberellin (GA) hormone signaling pathway, primarily functioning by regulating the activity of TFs in plants [51]. Therefore, it could be hypothesized that under saline-alkali stress, cotton plant stabilizes DELLA proteins by inhibiting active gibberellin levels and thus enhancing plant stress tolerance and adaptability.

Root hairs are crucial for nutrient acquisition and environmental interaction, and their elongation is of great significance for root development [52]. The plant hormone ethylene could induce root hair formation and elongation by promoting the accumulation of ROS in roots [53,54]. Nine DEGs involved in the ethylene signaling pathway were identified, which may influence ROS accumulation in cotton roots by regulating ethylene signaling, thereby reducing root tip damage under compound saline-alkali stress.

The MAPK signaling pathway is closely related to cell proliferation and differentiation [55]. Research indicated that key genes in the MAPK signaling pathway, such as *MKK4* and *MPK6*, can influence non-canonical auxin signaling and IDA peptide signaling pathways, thereby regulating the formation and growth of lateral roots [56]. Our study on cotton root tips also suggested that the MAPK signaling pathway might respond to compound saline-alkali stress by regulating non-canonical auxin signaling, hydrogen peroxide metabolism, and ethylene signaling pathways.

Additionally, 14 DEGs were identified in the cysteine and methionine metabolism pathways. These key DEGs might affect plant responses to compound saline-alkali stress by influencing pyruvate metabolism, sulfur metabolism, and the biosynthesis of terpenoid backbones. The cysteine and methionine metabolism pathways are crucial for sulfur metabolism and redox balance in organisms [57]. Studies have shown that cysteine and methionine are key metabolites related to sulfur (S) amino acids in the plant metabolic network, enhancing Arabidopsis tolerance to salt stress by balancing sulfur metabolism [58,59]. Other researchers have found that cysteine and methionine metabolism is closely related to soybean responses to abiotic stress, functioning through redox-based post-translational modifications (*PTM*) of proteins [60]. Our findings are highly consistent with these studies.

### 3.2. Role of Candidate Gene GhERF2 in Cotton’s Response to Compound Saline-Alkali Stress

*GhERF2* (*GH_A08G1918*) is a member of the *ERF* subfamily, which belongs to the *AP2*/*ERF* superfamily. In soybeans, *GmERFs* were found to enhance transgenic soybean tolerance to drought stress through interaction with *GmDREB1* [61]. *OsERF52* could directly regulate the expression of C-repeat binding factor (*CBF*) genes in rice, thereby enhancing cold tolerance [62]. Studies in *Arabidopsis* have shown that the promoter region of *AtERF71* contains regulatory elements involved in salt stress response, positively affecting salt tolerance [40]. *SlERF84* conferred ABA hypersensitivity to transgenic plants, enhancing tomato tolerance to drought and salt stress [63]. Notably, certain members of the *ERF* subfamily in *Krascheninnikovia arborescens* and *Xanthoceras sorbifolia* have been shown to be closely associated with salt-alkali stress [64,65]. *GsERF6* identified in soybean has been demonstrated to enhance tolerance of transgenic plants to salt-alkali stress [66]. These findings indicated that the *ERF* subfamily is widely studied as it is closely related to abiotic stresses such as salt, drought, cold, and saline-alkali.

Further VIGS investigation of *GhERF2* (*GH_A08G1918*) revealed that silenced plants displayed more severe wilting phenotypes, such as reduced fresh weight, decreased chlorophyll content, and lower SOD activity, along with higher malondialdehyde (MDA) accumulation. DAB staining showed that gene-silenced plants exhibited deeper staining under compound saline-alkali stress, indicating that silencing *GhERF2* led to more severe cellular damage and increased H_2_O_2_ accumulation under stress conditions. Our findings suggested that *GhERF2* might play a crucial role in the response of plants to compound saline-alkali stress in simulated secondary field environments.

## 4. Materials and Methods

### 4.1. Plant Materials and Growth Conditions

The cotton material used in this study was the genetic standard line TM-1 of upland cotton (TM-1 was provided by Dr. Fang Liu’s research group at the Institute of Cotton Research). The cotton planting was conducted in a greenhouse with an average humidity of 45–50% and a temperature of 28 ± 0.5 °C, under a 14-h light/10-h dark photoperiod. Cotton seeds were soaked in 3% hydrogen peroxide for 30 min for surface sterilization, then rinsed with deionized water and soaked in tap water for 12 h. Full and uniformly sized seeds were neatly arranged on moist filter paper, which was folded and placed at a certain angle in a plastic germination box. The germination box was kept in the dark, and 20 mL of tap water was added every two days. Five days later, the cotton seedlings were transferred to hydroponic boxes containing 6 L of modified Hoagland solution for further hydroponic cultivation. The complete nutrient solution (CK) consists of 2.5 mM KNO_3_, 2.5 mM Ca(NO_3_)_2_, 1 mM MgSO_4_, 0.5 mM (NH_4_)H_2_PO_4_, 0.1 mM FeNa-EDTA, and micronutrients (2 × 10^−4^ mM CuSO_4_, 1 × 10^−3^ mM ZnSO_4_, 2 × 10^−2^ mM H_3_BO_3_, 5 × 10^−6^ mM (NH_4_)6Mo7O_24_, and 1 × 10^−3^ mM MnSO_4_).

The hydroponic cotton seedlings obtained using the aforementioned germination method (two days after being transferred to hydroponic boxes) were used for transcriptome sequencing sample collection. The experimental group was treated with a combined salt-alkali solution for 20 min, while the control group was treated with distilled water for the same duration. After 20 min of salt-alkali stress treatment, cotton seedlings exhibited a certain degree of wilting. Therefore, samples were collected at this time point. Samples were then collected from root tips (with a standard length of 1 cm) and non-root tips of the cotton seedling roots. The experiment was performed in three biological replicates, each replicate consisting of 20 uniformly growing seedlings. All samples were rapidly frozen in liquid nitrogen and stored at −80 °C for subsequent transcriptome sequencing. The composition of the combined salt-alkali solution simulating the stress was designed based on the ionic composition of saline-alkali soil in Aral City, Tarim region, Xinjiang (Appendix A); the soil ion composition was tested at Anyang Institute of Technology.

Tobacco (*Nicotiana benthamiana*) was used as the wild type (WT) for transient transformation. Tobacco was also grown in a greenhouse with an average humidity of 60–70% and a temperature of 24 ± 0.5 °C, under a 14-h light/10-h dark photoperiod. Tobacco seeds were evenly sown in plastic seedling trays filled with a mixture of sand and vermiculite (in a 6:4 ratio), covered with plastic film for moisture retention, and transplanted into soil six days later. Tobacco plants were ready for subsequent transient transformation experiments after three weeks of growth.

### 4.2. RNA Extraction and Library Sequencing

Total RNAs were extracted using the RNA Prep Pure Plant Kit (Tiangen, Beijing, China). The RNA quality was assessed by agarose gel electrophoresis to check for degradation or contamination, and the RNA integrity was evaluated using the Nano 6000 Assay Kit of the Bioanalyzer 2100 system (Agilent Technologies, Santa Clara, CA, USA). After quantification with the Qubit^®^ RNA Assay Kit in a Qubit^®^ 2.0 Fluorometer (Life Technologies, Carlsbad, CA, USA), approximately 3 μg of RNA samples were used to prepare a cDNA library with the NEB-Next^®^ Ultra™ RNA Library Prep Kit for Illumina^®^ (NEB, Ipswich, MA, USA).

The 150–200 nt fragments were purified using the AMPure XP system (Beckman Coulter, Beverly, MA, USA) and then subjected to PCR amplification for enrichment and collection. Further purification of the PCR products was performed with the AMPure XP system, and the library quality was assessed on the Agilent Bioanalyzer 2100 system. The TruSeq Cluster Kit v3-cBot-HS (Illumina, San Diego, CA, USA) was used for clustering the index-coded samples with the cBot Cluster Generation System. A total of 12 cDNA libraries were sequenced on a flow cell using the Illumina HiSeq™ 2500 sequencing platform.

### 4.3. Transcriptome Sequencing and Data Analysis

After processing with in-house Perl scripts, the raw data in FASTQ was converted into read sequences, with corresponding base qualities recorded. Subsequently, low-quality reads (those with adapters, poly-N > 10% or Q20 < 20%) were filtered out to obtain clean data, from which the Q20 and GC contents were calculated. Alignment was conducted using TopHat v2.0.12 software with clean reads aligned to the *G. hirsutum* TM-1 reference genome [66]. Genome data and gene model annotation files were obtained from the CottonGen database (https://www.cottongen.org/species/Gossypium_hirsutum/UTX-TM1_v2.1, accessed on 11 May 2024). Conduct PCA analysis using OmicShare (https://www.omicshare.com/tools/, accessed on 20 May 2024). Perform PCC analysis using omicstudio (https://www.omicstudio.cn/tool, accessed on 20 May 2024). Expression levels of the genes were assessed by counting read numbers mapped to each gene using HTSeq v0.6.1 [67], and FPKM was utilized for quantifying gene expression levels based on gene length and mapped read counts. Differential gene expression analysis was performed using DESeq2 between two different groups (and by the edgeR package in R version 4.4.0 for comparisons between two samples) [68], with differentially expressed genes/transcripts set as the values of false discovery rate (FDR) less than 0.05 and absolute fold change equal to and greater than 1.5. The functional enrichment of KEGG pathways and GO categories was analyzed using OmicShare (https://www.omicshare.com/tools/, accessed on 23 May 2024).

### 4.4. qRT-PCR Analyses of Gene Expression

In order to verify the authenticity and reliability of this transcriptome sequencing data, a quantitative real-time PCR (qRT-PCR) experiment was conducted on 20 randomly selected DEGs with three biological and three technical replicates per sample. Specific primers were designed using the Primer-BLAST online tool on the NCBI website, and the primer sequences are detailed in Appendix A. The extraction method for cotton seedling roots was consistent with the RNA extraction and library sequencing procedures mentioned above. cDNA synthesis was performed using the FastKing one-step genomic cDNA first-strand synthesis premix reagent (TIANGEN, Beijing, China). The ChamQ Universal SYBR qPCR Master Mix (Vazyme, Nanjing, China) was utilized for the qRT-PCR experiment with the synthesized cDNA as a template. The cotton gene *GhUBQ7* (DQ116441) was used as an internal reference gene, and the expression levels of DEGs were calculated using the 2^−∆∆CT^ method [69].

### 4.5. Cloning of GhERF2 and Transient Transformation in Tobacco

Using cDNA extracted and reverse-transcribed from the roots of TM-1 as a template (the RNA extraction and reverse transcription kits were the same as those used in the aforementioned qRT-PCR experiments), the DNA sequence of the open reading frame (ORF) of *GhERF2* was downloaded from Cotton FGD (https://cottonfgd.net/, accessed on 30 May 2024). Primers were designed using SnapGene to amplify the CDS excluding the terminator. The plasmid pCMBIA2300-DsRED2:GhERF2 was constructed and transformed into Agrobacterium. Agrobacterium was injected into tobacco leaves, and after 48 h of incubation in the dark, the leaves were observed under a laser confocal microscope.

### 4.6. Virus-Induced Gene Silencing (VIGS) Experiment

The TRV2 (tobacco rattle virus 2) vector was used for the VIGS experiment, and the stability of the VIGS system was validated using the albino gene *GhCLA1* [70]. Both systems were preserved in our laboratory. Specific primers were designed using the SGN-VIGS website (https://vigs.solgenomics.net/, accessed on 17 June 2024) to amplify the fragment of *GhERF2* and insert it into the binary vector pYL156 of the tobacco rattle virus (*TRV2*) via the endonucleases *EcoRI* and *KpnI* to construct the plasmid TRV:GhERF2, which was transformed into Agrobacterium. Agrobacterium was injected into the abaxial surface of fully expanded cotyledons of cotton seedlings, and true leaves were collected after the appearance of the albino phenotype to assess silencing efficiency. Details of the *GhERF2* vector primer sequences designed in this study are provided in Appendix A.

## 5. Conclusions

Developing and promoting saline-alkali-resistant varieties is one of the best solutions to achieve sustainable land development in the context of intensified soil salinization. Therefore, it is crucial to understand the molecular mechanisms of plant responses to saline-alkali stress and identify key genes for further developing superior saline-alkali-resistant varieties. In this study, transcriptome sequencing was performed on root samples of seedlings under simulated secondary field stress incurred by salt and alkali complexes. Enrichment analysis revealed that DEGs were significantly enriched in plant hormone signal transduction, MAPK signaling pathways, and cysteine and methionine metabolism pathways. The expression of the key TF gene *GH_A08G1918* (*GhERF2*), which is closely related to the ethylene and MAPK signaling pathways, was significantly induced by compound saline-alkali stress. Silencing this gene via VIGS in the upland cotton line TM-1 resulted in more severe wilting phenotypes. DAB staining further supported the positive regulatory role of *GhERF2* in cotton’s response to saline-alkali stress. These findings provide valuable insights for advancing research on cotton’s saline-alkali resistance, enhancing our understanding of the mechanisms underlying cotton’s saline-alkali tolerance, and offering new resistant materials for breeding.

## Figures and Tables

**Figure 1 plants-14-00756-f001:**
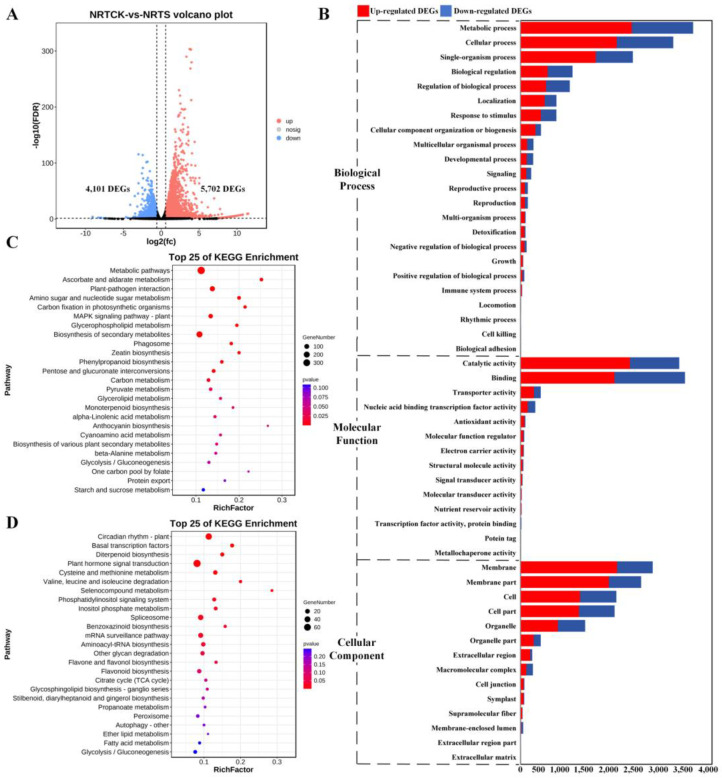
Identification and functional enrichment analysis of DEGs between NRTCK and NRTS. (**A**) Volcano plot of DEGs from two pairwise comparisons, with FDR < 0.05 and |log2(fold-change)| > 1.5. (**B**) Enriched GO terms of DEGs from two pairwise comparisons. (**C**) Enriched KEGG pathways of up-regulated DEGs from two pairwise comparisons. (**D**) Enriched KEGG pathways of down-regulated DEGs from two pairwise comparisons.

**Figure 2 plants-14-00756-f002:**
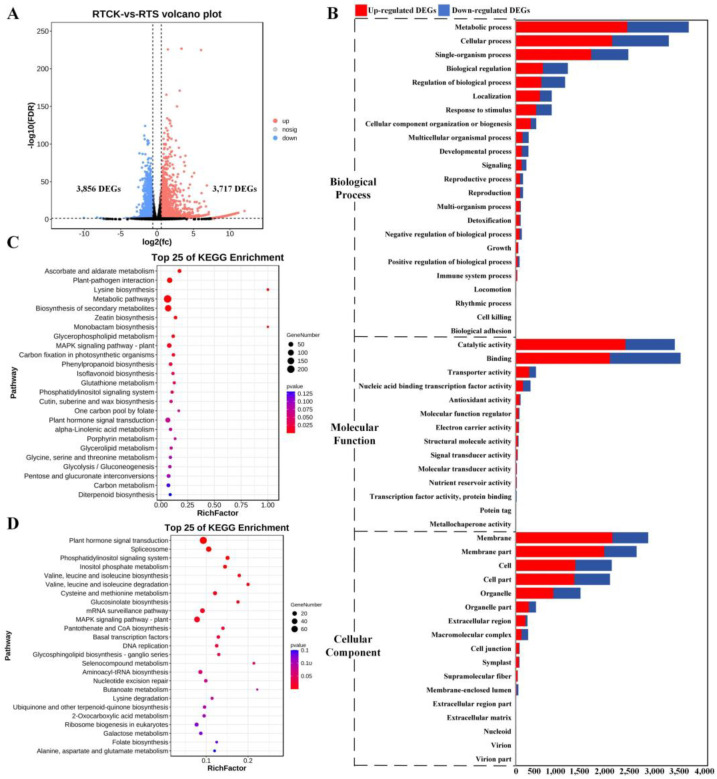
Identification and functional enrichment analysis of DEGs between RTCK and RTS. (**A**) Volcano plot of DEGs from two pairwise comparisons, with FDR < 0.05 and |log2(fold-change)| > 1.5. (**B**) Enriched GO terms of DEGs from two pairwise comparisons. (**C**) Enriched KEGG pathways of up-regulated DEGs from two pairwise comparisons. (**D**) Enriched KEGG pathways of down-regulated DEGs from two pairwise comparisons.

**Figure 3 plants-14-00756-f003:**
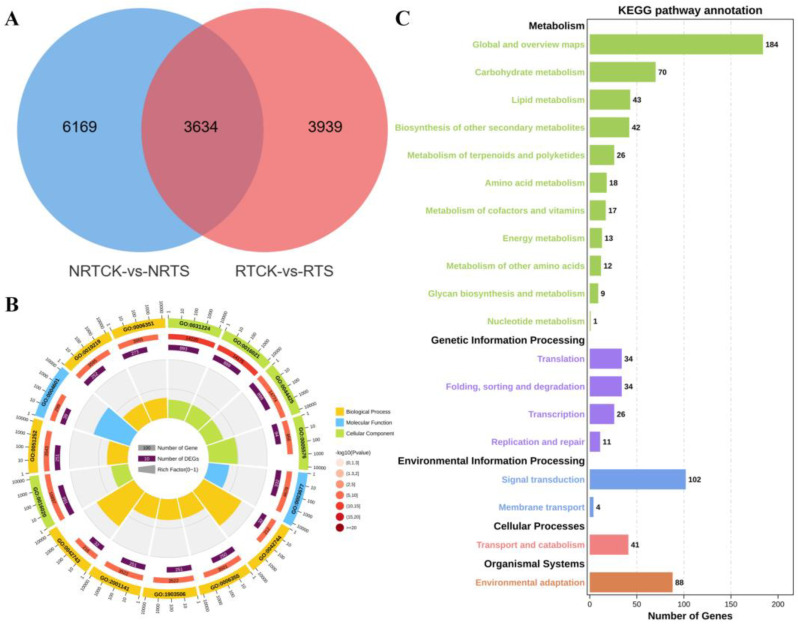
Identification and functional enrichment analysis of common DEGs. (**A**) Venn diagram of DEGs obtained from root tip and non-root tip regions, including all up-regulated and down-regulated DEGs. (**B**) Enriched GO terms of DEGs from two pairwise comparisons. (**C**) Enriched KEGG pathways of DEGs from two pairwise comparisons.

**Figure 4 plants-14-00756-f004:**
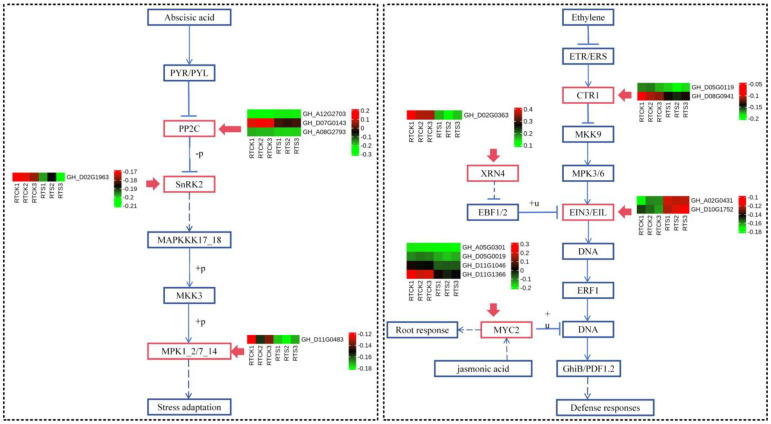
Heat-map of DEGs participating in the pathways of abscisic acid and ethylene. The red boxes represent the DEGs from the root tip.The solid blue arrows represent single-step biochemical reactions, while the dashed blue arrows indicate indirect or uncertain biochemical reactions.

**Figure 5 plants-14-00756-f005:**
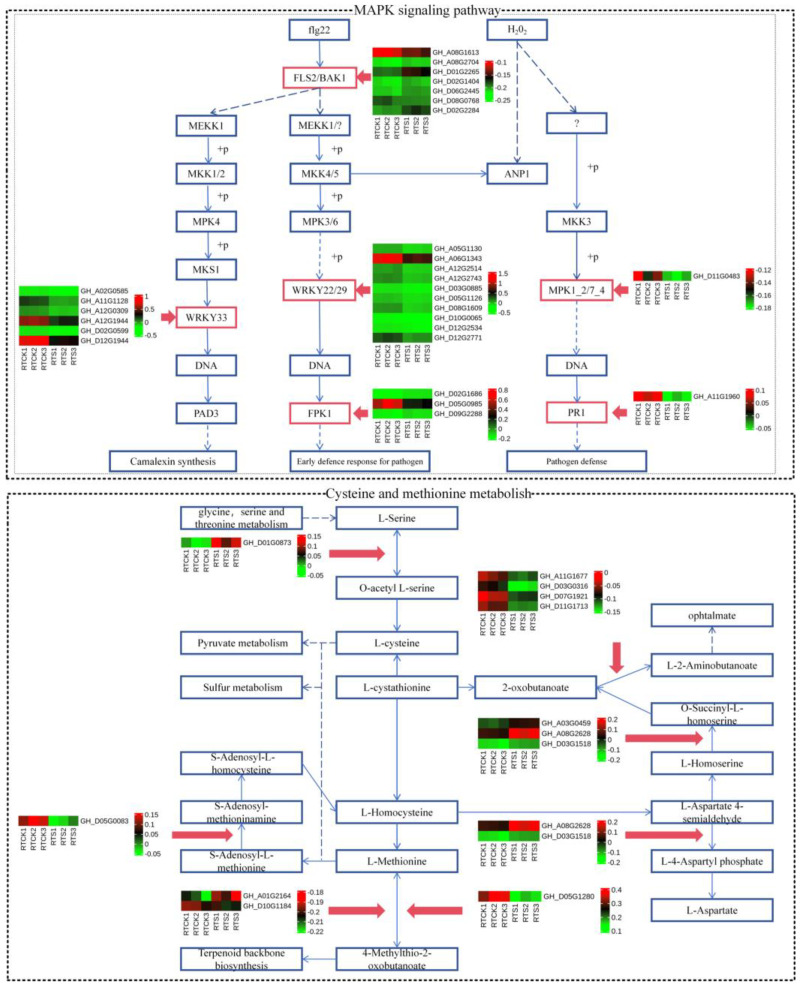
Heat-map of DEGs participating in the MAPK signaling pathways and cysteine and methionine metabolism. The red boxes represent the DEGs from the root tip.The solid blue arrows represent single-step biochemical reactions, while the dashed blue arrows indicate indirect or uncertain biochemical reactions.

**Figure 6 plants-14-00756-f006:**
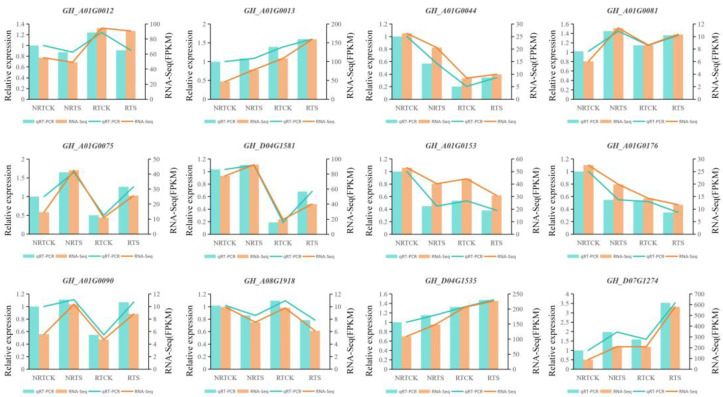
Quantitative real-time PCR (qRt-PCR) verification and RAN-seq data of 12 Randomly selected DEGs.

**Figure 7 plants-14-00756-f007:**
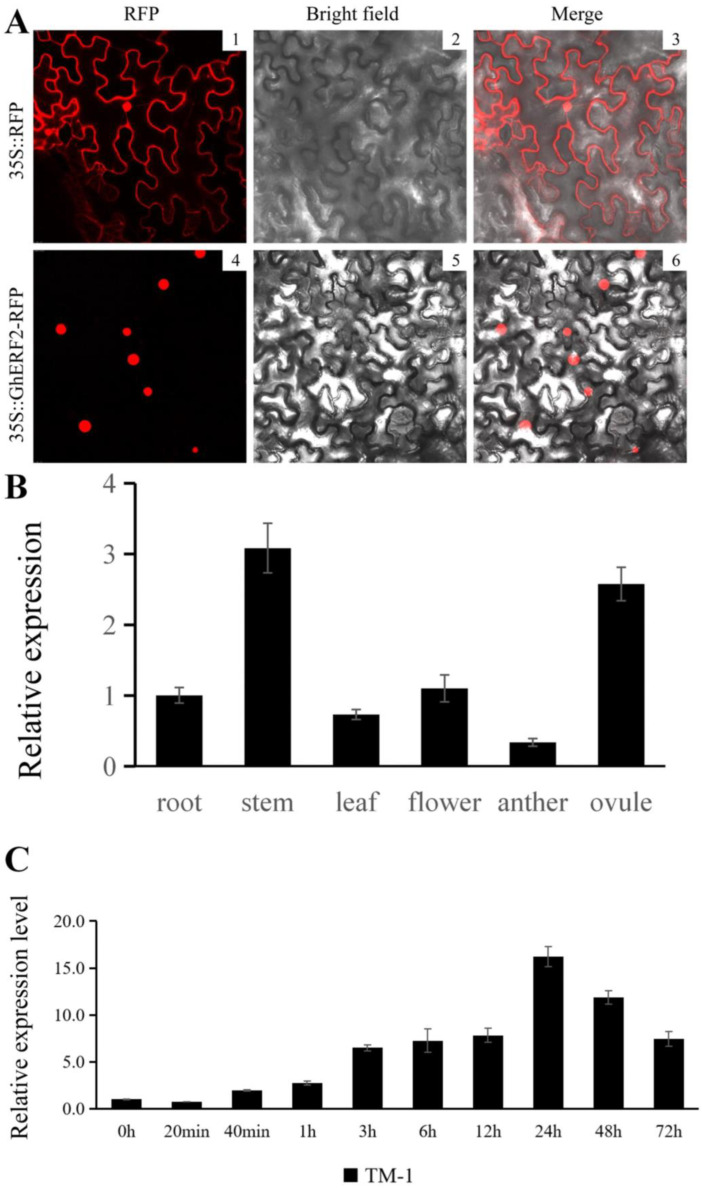
Expression pattern of *GhERF2*. (**A**) Subcellular localization of *GhERF2* protein in leaf epidermal cells using the *Nicotiana benthamiana* transient expression system. The *35S::RFP* vector without any *GhERF2* sequence served as a control. 1–3: RFP fluorescence detection in tobacco leaves transformed with *35S*::*RFP*; 4–6: RFP fluorescence detection in tobacco leaves transformed with 35S::*GhERF2*-*RFP*; 1 and 4: RFP fluorescence; 2 and 5: bright-field images; 3 and 6: merged images. The red section represents RFP fluorescence. (**B**) Expression levels of *GhERF2* in different cotton tissues determined by the qRT-PCR experiment. (**C**) Expression levels of *GhERF2* at various time points following compound saline-alkali stress in cotton root, as determined by qRT-PCR. All experiments included three biological and technical replicates, yielding consistent results.

**Figure 8 plants-14-00756-f008:**
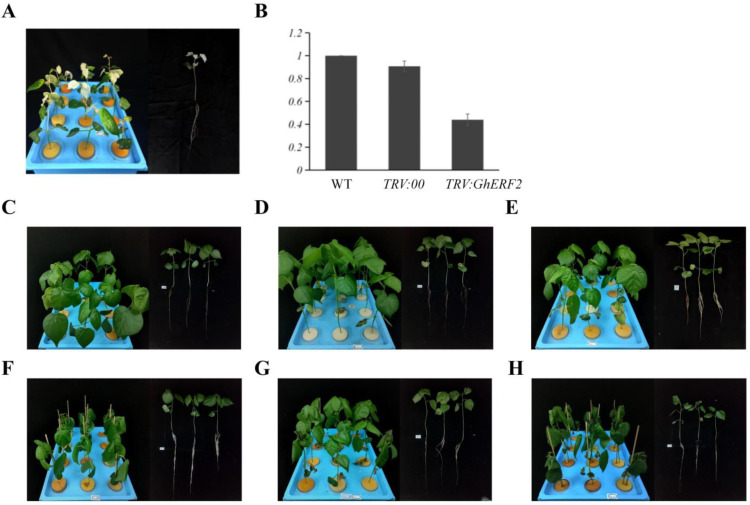
Silencing of *GhERF2* via Virus-Induced Gene Silencing (VIGS) increases cotton sensitivity to compound saline-alkali stress. (**A**) Phenotype of *VIGS*-*GhCLA1* plants under mock conditions (including whole pot and individual plant). (**B**) Expression levels of *GhERF2* in WT, *TRV*:*00*, and *TRV*:*GhERF2* determined by qRT-PCR. The cotton gene *GhUBQ7* (DQ116441) was used as an internal reference gene. (**C**) Phenotype of WT wild-type TM-1 under mock conditions (including whole pot and individual plant). (**D**) Phenotype of *TRV*:00 under mock conditions (including whole pot and individual plant). (**E**) Phenotype of *TRV*:*GhERF2* under mock conditions (including whole pot and individual plant). (**F**) Phenotype of WT wild-type TM-1 under compound saline-alkali conditions (including whole pot and individual plant). (**G**) Phenotype of *TRV*:*00* plants under compound saline-alkali conditions (including whole pot and individual plant). (**H**) Phenotype of *TRV*:*GhERF2* plants under compound saline-alkali conditions (including whole pot and individual plant).

**Figure 9 plants-14-00756-f009:**
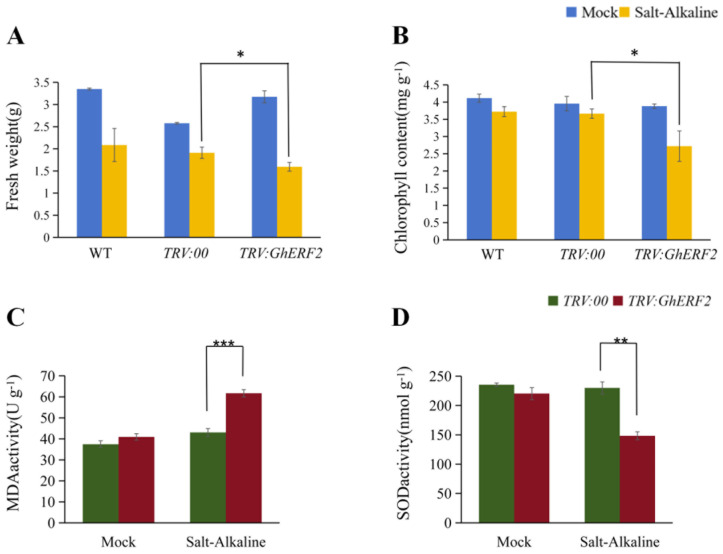
Physiological and biochemical measurements of VIGS plants. (**A**) Whole plant fresh weight. (**B**) Chlorophyll content. (**C**) MDA content. (**D**) SOD activity. *, **, and *** indicate significant differences at the 0.05, 0.01, and 0.001 levels, respectively. All experiments included at least three biological replicates.

**Figure 10 plants-14-00756-f010:**
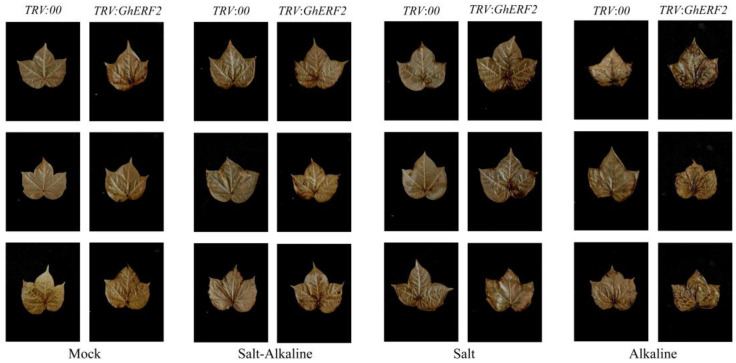
Leaf staining of transgenic and control plants, observing hydrogen peroxide levels via DAB staining. Three plants with similar growth were randomly selected from each treatment group as biological replicates.

**Table 1 plants-14-00756-t001:** RNA-seq data and quality assessment.

Sample Name	Clean Reads	Clean Base	Read Length	Q20 (%)	GC (%)
NRCK1	24,013,230	7,203,969,000	PE150	96.30	43.93
NRCK2	24,069,156	7,220,746,800	PE150	96.56	43.98
NRCK3	24,102,895	7,230,868,500	PE150	96.01	43.94
RCK1	23,546,010	7,063,803,000	PE150	96.19	44.27
RCK2	24,006,143	7,201,842,900	PE150	96.45	44.18
RCK3	24,115,835	7,234,750,500	PE150	96.30	44.07
RS1	24,097,237	7,229,171,100	PE150	96.53	44.43
RS2	24,130,131	7,239,039,300	PE150	96.54	44.41
RS3	24,044,474	7,213,342,200	PE150	96.59	44.49
NRS1	24,073,680	7,222,104,000	PE150	96.58	44.70
NRS2	24,138,006	7,241,401,800	PE150	96.72	44.73
NRS3	24,114,157	7,234,247,100	PE150	96.62	44.67

## Data Availability

The raw data supporting this study can be found on the Genome Sequence Archive (GSA) website (https://bigd.big.ac.cn/gsa/browse/CRA022812, accessed on 11 February 2025) and its Appendix A.

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
