# Peer review of "Tissue-Specific RNA-Seq Analysis of Cotton Roots’ Response to Compound Saline-Alkali Stress and the Functional Validation of the Key Gene GhERF2"

_plants, 2025, doi:10.3390/plants14050756_

Round 1

Reviewer 1 Report

Comments and Suggestions for Authors

The research article under consideration addresses the impact of alkali-saline stress conditions on cotton root tissues. The impact of eustress and distress conditions on plant or animal cell signal cascades has been demonstrated to lead to serious physiological processes. The investigation of such processes is imperative for a more profound comprehension of cell adaptation mechanisms against pathological stressors. In this regard, the subject of the present manuscript is of particular significance to the field.

The authors of the study detected and evaluated relevant markers of stress. Especially, their focus was on the concentrations of malondialdehyde and hydrogen peroxide. The conventional methods were employed in this study. Nevertheless, it is not advisable to employ diaminobenzidine for hydrogen peroxide detection in this case. The amino groups of diaminobenzidine are susceptible to oxidation by molecular oxygen present in tissue samples. In this process, molecular oxygen undergoes a reduction, resulting in the formation of either superoxide anion radicals or hydrogen peroxide. Consequently, the concentrations of hydrogen peroxide can be artificially elevated.

Nevertheless, the authors have compiled a substantial body of experimental data that substantiates the findings and conclusions of the study. The experimental data is meticulously documented and presented in the form of illustrative figures and tables. Additionally, the discussion section is meticulously crafted, providing a comprehensive exploration of the subject matter. The authors have cited the most relevant references.

Author Response

Dear Editor and reviewers:

    On behalf of my co-authors, we thank you very much for your helpful efforts processing our manuscript entitled " Tissue-specific RNA-seq analysis of cotton roots response to compound saline-alkali stress and the functional validation of the key gene GhERF2" and providing us an opportunity to revise it. With regard to your and the reviewers’ positive and constructive comments on our manuscript, we think they are of high value and importance for improving our manuscript, as well as of critical guiding significance to our future researches.

After careful reviewing on your comments, we have made correspondent revisions in the manuscript, which we hope will meet the requirements of your journal. We have also sought for a professional help from a native specialist to improve the logicality and readability of our manuscript. A response to the reviewers’ comments is attached hereinafter for your review.

Thank you again for your kind help and efforts. If there is anything that need us do, please feel free to let us know.

Best regards, 

Pengtao Li

E-mail: lipengtao1056@126.com

Attachments:

Response to the comments of Reviewer 1

  1. Comment:The research article under consideration addresses the impact of alkali-saline stress conditions on cotton root tissues. The impact of eustress and distress conditions on plant or animal cell signal cascades has been demonstrated to lead to serious physiological processes. The investigation of such processes is imperative for a more profound comprehension of cell adaptation mechanisms against pathological stressors. In this regard, the subject of the present manuscript is of particular significance to the field.

The authors of the study detected and evaluated relevant markers of stress. Especially, their focus was on the concentrations of malondialdehyde and hydrogen peroxide. The conventional methods were employed in this study. Nevertheless, it is not advisable to employ diaminobenzidine for hydrogen peroxide detection in this case. The amino groups of diaminobenzidine are susceptible to oxidation by molecular oxygen present in tissue samples. In this process, molecular oxygen undergoes a reduction, resulting in the formation of either superoxide anion radicals or hydrogen peroxide. Consequently, the concentrations of hydrogen peroxide can be artificially elevated.

Nevertheless, the authors have compiled a substantial body of experimental data that substantiates the findings and conclusions of the study. The experimental data is meticulously documented and presented in the form of illustrative figures and tables. Additionally, the discussion section is meticulously crafted, providing a comprehensive exploration of the subject matter. The authors have cited the most relevant references.

Response: Thank you very much for your valuable comments. We totally agree with your concerns regarding the hydrogen peroxide detection. However, we have performed the experiments by standardizing experimental operations and optimizing experimental procedures to minimize the potential impact of artificial factors in the DAB staining process. We have also performed a series of experiments, the results of which can support each other. Therefore, we believe our results are reliable on these bases.

Reviewer 2 Report

Comments and Suggestions for Authors

Review Plants 347511

            Breeding of saline-alkali-resistant varieties is one of the best solutions to overcome soil salinization Therefore, the study of Zhang and colleagues aimed to understanding of the transcriptional regulation of saline-alkali resistance genes in cotton and the prospects for application of GhERF2, a member of the ethylene response factor of ERF/AP2 transcription factor family could provide valuable information for further study of resistance to saline and alkali in cotton.

The manuscript submitted for publication in the journal includes two unequal parts. In the first, the authors presented the results of transcriptome sequencing of two of different root regions after exposure to compound saline-alkali stress for 20 minutes. The authors do not provide any arguments for the choice of such an experimental model, especially since in the second part of the research the expression levels of GhERF2 in different cotton tissues determined by qRt-PCR experiment reached its peak at 24h. Although the authors meticulously list functional enrichment of the tissue-specific DEGs and their top KEGG pathways, they refrain from further analysis of their possible impact on salt tolerance. Since the paper does not provide the original data of the transcriptome analysis (Supplement is missing), the reader must take it for granted that this process involves mainly plant hormone signal transduction, MAPK signaling pathway-, and cysteine and methionine metabolism. While assessing hormonal regulation, the authors mention genes related to abscisic acid, ethylene, brassinosteroids and gibberellins, but they only provide data for DEGs participating in the ABA and ethylene pathways without analyzing the reaction of target genes. The carelessness of the evaluation of the results is evidenced by Figure 3, where the data provided does not correspond to the headings, or is completely absent ((A) Principal component analysis of the 12 samples). Quantitative qRt-PCR verification does not carry an additional meaning, since it does not consider the functions of the analyzed genes.

The second part of the research is more logical. The authors convincingly demonstrate that silencing of GhERF2 in cotton line TM-1 results in severe wilting phenotypes, thus supporting the positive regulatory role of this gene in cotton's response to saline-alkali stress. It should be noted however, that DAB staining reveals only hydrogen peroxide. O2- levels (lines 403,407) are visualized with NBT staining that is missing in the study.

i

Comments on the Quality of English Language

The English should  be improved

Author Response

Dear Editor and reviewers:

    On behalf of my co-authors, we thank you very much for your helpful efforts processing our manuscript entitled " Tissue-specific RNA-seq analysis of cotton roots response to compound saline-alkali stress and the functional validation of the key gene GhERF2" and providing us an opportunity to revise it. With regard to your and the reviewers’ positive and constructive comments on our manuscript, we think they are of high value and importance for improving our manuscript, as well as of critical guiding significance to our future researches.

After careful reviewing on your comments, we have made correspondent revisions in the manuscript, which we hope will meet the requirements of your journal. We have also sought for a professional help from a native specialist to improve the logicality and readability of our manuscript. A response to the reviewers’ comments is attached hereinafter for your review.

Thank you again for your kind help and efforts. If there is anything that need us do, please feel free to let us know.

Best regards, 

Pengtao Li

E-mail: lipengtao1056@126.com

Attachments:

Response to the comments of Reviewer 2 

  1. Comment:The authors do not provide any arguments for the choice of such an experimental model, especially since in the second part of the research the expression levels of GhERF2in different cotton tissues determined by qRt-PCR experiment reached its peak at 24h. Although the authors meticulously list functional enrichment of the tissue-specific DEGs and their top KEGG pathways, they refrain from further analysis of their possible impact on salt tolerance.

Response: Thank you very much for your valuable comments. In the previous study, we perform single-cell RNA-seq experiment on cotton roots in response to salt stress, resulting in that root tips might be the specific tissues or ports or that the most positively respond to adversity stresses (Li P, Liu Q, Wei Y, Xing C, Xu Z, Ding F, Liu Y, Lu Q, Hu N, Wang T, Zhu X, Cheng S, Li Z, Zhao Z, Li Y, Han J, Cai X, Zhou Z, Wang K, Zhang B, Liu F, Jin S, Peng R. Transcriptional Landscape of Cotton Roots in Response to Salt Stress at Single-cell Resolution. Plant Commun. 2023, 5(2):100740. doi: 10.1016/j.xplc.2023.100740.). After the preliminary experiments on the cotton roots under saline-alkali stress, we found the compound saline-alkali stress could result in the more serious damages on the plant development. Besides, under less than 20 mins saline-alkali stress, cotton seedlings exhibited a certain degree of wilting, of which cotton tips could be utilized for further single-cell RNA-seq experiments. Therefore, the different root tissues were designed in this study to check whether the above-mentioned inference is right or not. Obviously, the candidate gene GhERF2 was screened based on the pairwise comparisons between the samples, and its potential functions were also verified by VIGS on the cotton plants under saline-alkali stress according to the common practice (Mu C, Zhou L, Shan L, Li F, Li Z. Phosphatase GhDsPTP3a interacts with annexin protein GhANN8b to reversely regulate salt tolerance in cotton (Gossypium spp.). New Phytol. 2019, 223(4):1856-1872. doi: 10.1111/nph.15850.).

  1. Comment:Since the paper does not provide the original data of the transcriptome analysis (Supplement is missing), the reader must take it for granted that this process involves mainly plant hormone signal transduction, MAPK signaling pathway-, and cysteine and methionine metabolism.

Response: Thank you very much for your precious comments. We have provided the raw data for transcriptome analysis, which are still under check.

  1. Comment:While assessing hormonal regulation, the authors mention genes related to abscisic acid, ethylene, brassinosteroids and gibberellins, but they only provide data for DEGs participating in the ABA and ethylene pathways without analyzing the reaction of target genes.

Response: Thank you very much for your valuable feedback. We have redrawn the pathway diagram for DEGs related to the two pathways of brassinosteroids and gibberellins. Please refer to Supplementary Figure S3

  1. 4.Comment:The carelessness of the evaluation of the results is evidenced by Figure 3, where the data provided does not correspond to the headings, or is completely absent ((A) Principal component analysis of the 12 samples).

Response: Thank you very much for your great comments. We changed the title of Figure 3. Please refer to lines 254-258: Figure 3. Identification and Functional Enrichment Analysis of DEGs (A) Venn diagram of DEGs obtained from root tip and non-root tip regions, including all up-regulated and down-regulated DEGs.(B) The enriched GO terms of DEGs from two pairwise comparisons. (C) The enriched KEGG pathways of DEGs from two pairwise comparisons.

  1. 5.Comment:The second part of the research is more logical. The authors convincingly demonstrate that silencing of GhERF2 in cotton line TM-1 results in severe wilting phenotypes, thus supporting the positive regulatory role of this gene in cotton's response to saline-alkali stress. It should be noted however, that DAB staining reveals only hydrogen peroxide. O2- levels (lines 403,407) are visualized with NBT staining that is missing in the study.

Response: Thank you very much for your great comments. We have revised the corresponding parts of the manuscript according to your suggestions. Please refer to lines 399-403: Leaf staining with 3,3'-diaminobenzidine (DAB), which was also used to detect hydrogen peroxide levels, revealed that hydrogen peroxide levels in TRV:GhERF2 plant leaves were significantly higher than that in TRV:00 plant leaves (Figure 10). These results suggested that the ability of GhERF2-silenced plants to scavenge ROS was reduced

Reviewer 3 Report

Comments and Suggestions for Authors

This manuscript aims to characterize saline-alkali stress response in cotton roots. The analysis is mostly solid while there are still some missing information.  Overall, the focus is not clear, I have doubt on whether MAPK is critical in saline stress, especially in root. In addition, authors need to link different parts of data in a logical fashion, instead of dumping a large amount of data. 

1.       Please, identify the source of plant samples (e.g. sources of seed, vendor, particular treatment) and confirm they are the correct samples.

2.       Several references are missing (e.g. HTSeq1, KEGG, DESeq2).   

3.       Supplementary information is not provided.

4.       Figure 3: It is not clear which comparison was used for Figure 3C. Also, the main text does not seem to match Figure 3C. Figure 3D is missing.

5.       Section 2.4: It is not clear which DEGs were used (tip, non-tip, union or intersection).  The number of DEGs are very large in this study.  So, it is not surprising authors can pick out DEGs in almost any plant associated pathways.  Authors need to point that DEGs are specifically enriched in each pathway (fold enrichment, or p value).   “These findings suggested that root tips respond to saline-alkali stress through multiple plant hormone pathways, such as inhibiting the gib-berellin pathway.’ (lines 265-267) seems to be an overstatement with the current structure. 

6.       Section 2.6: It is not clear why authors chose to use GhERF2.  RNA-seq does not seem to support any particular function of it (at least with the given data).  The abstract sounds like authors found an unexpected role of GhERF2, but I am not sure why. Authors needs to show shy ERF2 maybe more important (or more sticking out) than other MAPK genes. 

7.       Figure 7C: Which tissue was used?  Is this associated with roots?

8.       Figure 8 (currently marked as Fig 6).  Pictures are extremely qualitative, and turning these into some graph (like bargraphs) would be helpful. 

9.       One of the major drawback is the effect of MAPK pathway in saline-alkali stress. In general, it is not surprising to find MAPK in stress response because it is associated with so many processes in cell.  In addition, it is not clear if the effect is specific to roots given the data from this manuscript.  The manuscript would be stronger if authors can demonstrate that these found effect is saline-specific and root-specific.

Author Response

Dear Editor and reviewers:

    On behalf of my co-authors, we thank you very much for your helpful efforts processing our manuscript entitled " Tissue-specific RNA-seq analysis of cotton roots response to compound saline-alkali stress and the functional validation of the key gene GhERF2" and providing us an opportunity to revise it. With regard to your and the reviewers’ positive and constructive comments on our manuscript, we think they are of high value and importance for improving our manuscript, as well as of critical guiding significance to our future researches.

After careful reviewing on your comments, we have made correspondent revisions in the manuscript, which we hope will meet the requirements of your journal. We have also sought for a professional help from a native specialist to improve the logicality and readability of our manuscript. A response to the reviewers’ comments is attached hereinafter for your review.

Thank you again for your kind help and efforts. If there is anything that need us do, please feel free to let us know.

Best regards, 

Pengtao Li

E-mail: lipengtao1056@126.com

Attachments:

Response to the comments of Reviewer 3

  1. Comment:Please, identify the source of plant samples (e.g. sources of seed, vendor, particular treatment) and confirm they are the correct samples.

Response: Thank you very much for your valuable comments. We have added relevant information to the manuscript. Please refer to lines 536-538: The cotton material used in this study was the genetic standard line TM-1 of upland cotton (TM-1 was provided by Dr. Fang Liu’s research group at the Institute of Cotton Research).

  1. Comment:Several references are missing (e.g. HTSeq1, KEGG, DESeq2).

Response: Thank you for your precious comments. We have added the relevant references in the manuscript. Please refer to lines 594-604: Expression levels of the genes were assessed by counting read numbers mapped to each gene using HTSeq v0.6.1 [68], and FPKM was utilized for quantifying gene expression levels based on gene length and mapped read counts. Differential gene expression analysis was performed using DESeq2 between two different groups (and by the edgeR package in R version 4.4.0 for comparisons between two samples)[69], with differentially expressed genes/transcripts were set as the values of false discovery rate (FDR) less than 0.05 and absolute fold change equal to and more than 1.5. The functional enrichment of KEGG pathways and GO categories were analyzed using OmicShare (https://www.omicshare.com/tools/, accessed on 23 May 2024).

  1. Comment:Supplementary information is not provided.

Response: Thank you very much for your valuable feedback. We have re-provided supplementary materials related to the manuscript.

  1. 4.Comment:Figure 3: It is not clear which comparison was used for Figure 3C. Also, the main text does not seem to match Figure 3C. Figure 3D is missing.

Response: Thank you very much for your great commnets. We changed the title of Figure 3. Please refer to lines 258-264: Figure 3. Identification and Functional Enrichment Analysis of DEGs (A) Venn diagram of DEGs obtained from root tip and non-root tip regions, including all up-regulated and down-regulated DEGs.(B) The enriched GO terms of DEGs from two pairwise comparisons. (C) The enriched KEGG pathways of DEGs from two pairwise comparisons.

  1. 5.Comment:Section 2.4: It is not clear which DEGs were used (tip, non-tip, union or intersection). The number of DEGs are very large in this study. So, it is not surprising authors can pick out DEGs in almost any plant associated pathways. Authors need to point that DEGs are specifically enriched in each pathway (fold enrichment, or p value). "These findings suggested that root tips respond to saline-alkali stress through multiple plant hormone pathways, such as inhibiting the gib-berellin pathway." (lines 265-267) seems to be an overstatement with the current structure.

Response: Thank you very much for your great comments. We have revised the corresponding parts of the manuscript according to your suggestions. Please refer to lines 206-270: Through functional enrichment analysis on DEGs specifically expressed in root tips, it was preliminarily suggested that plant hormone signal transduction, MAPK signaling pathway-plant, and cysteine and methionine metabolism were associated with compound saline-alkali stress. A total of 74 DEGs were identified within the eight key pathways of the plant hormone signal transduction (Figure 4 and Supplementary Figure S3), including 5 DEGs related to abscisic acid (all down-regulated ones), 9 DEGs to ethylene (2 up-regulated and 7 down-regulated ones), 23 DEGs to auxin (6 up-regulated and 17 down-regulated ones), 9 DEGs to brassinosteroid (5 up-regulated and 4 down-regulated ones), and 22 DEGs to gibberellin (all the down-regulated ones). These findings implied that root tips respond to saline-alkali stress through multiple plant hormone pathways.. We have deleted "These findings suggested that root tips respond to saline-alkali stress through multiple plant hormone pathways, such as inhibiting the gibberellin pathway." and revised our description.

  1. 6.Comment:Section 2.6: It is not clear why authors chose to use GhERF2. RNA-seq does not seem to support any particular function of it (at least with the given data). The abstract sounds like authors found an unexpected role of GhERF2, but I am not sure why. Authors needs to show shy ERF2 maybe more important (or more sticking out) than other MAPK genes.

Response: Thank you very much for your valuable advice. In the previous study, we perform single-cell RNA-seq experiment on cotton roots in response to salt stress, resulting in that root tips might be the specific tissues or ports or that the most positively respond to adversity stresses (Li P, Liu Q, Wei Y, Xing C, Xu Z, Ding F, Liu Y, Lu Q, Hu N, Wang T, Zhu X, Cheng S, Li Z, Zhao Z, Li Y, Han J, Cai X, Zhou Z, Wang K, Zhang B, Liu F, Jin S, Peng R. Transcriptional Landscape of Cotton Roots in Response to Salt Stress at Single-cell Resolution. Plant Commun. 2023, 5(2):100740. doi: 10.1016/j.xplc.2023.100740.). Besides, we found that the GhERF2 could positively respond both salt and saline-alkali stresses in cotton root tips based on the single-cell RNA-seq and bulked RNA-seq (Liu Q, Li P, Cheng S, Zhao Z, Liu Y, Wei Y, Lu Q, Han J, Cai X, Zhou Z, Umer MJ, Peng R, Zhang B, Liu F. Protoplast Dissociation and Transcriptome Analysis Provides Insights to Salt Stress Response in Cotton. Int J Mol Sci. 2022, 23(5):2845. doi: 10.3390/ijms23052845.), which motived us to chose this gene as the potential candidate gene for further function verification. Obviously, the tissue-specific transcriptome experiments are also consistent with our speculation.

  1. 7.Comment:Figure 7C: Which tissue was used?  Is this associated with roots?

Response: Thank you very much for your careful and valuable reviewing. We have revised the corresponding parts of the manuscript according to your suggestions. Please refer to lines 333-335:(C) Expression levels of GhERF2 in cotton root at various time points after compound saline-alkali stress via qRT-PCR. All experiments included three biological and technical replicates, yielding consistent results.

  1. 8.Comment:Figure 8 (currently as Fig 6). Pictures are extremely qualitative, and turning these into some graph (like bargraphs) would be helpful.

Response: Thank you very much for your worthy suggestion. We have made modifications to the relevant part and renamed it as Figure 6

  1. 9.Comment:One of the major drawback is the effect of MAPK pathway in saline-alkali stress. In general, it is not surprising to find MAPK in stress response because it is associated with so many processes in cell.  In addition, it is not clear if the effect is specific to roots given the data from this manuscript.  The manuscript would be stronger if authors can demonstrate that these found effect is saline-specific and root-specific.

Response: Thank you very much for your careful and valuable reviewing. In the previous study, we perform single-cell RNA-seq experiment on cotton roots in response to salt stress, resulting in that root tips might be the specific tissues or ports or that the most positively respond to adversity stresses (Li P, Liu Q, Wei Y, Xing C, Xu Z, Ding F, Liu Y, Lu Q, Hu N, Wang T, Zhu X, Cheng S, Li Z, Zhao Z, Li Y, Han J, Cai X, Zhou Z, Wang K, Zhang B, Liu F, Jin S, Peng R. Transcriptional Landscape of Cotton Roots in Response to Salt Stress at Single-cell Resolution. Plant Commun. 2023, 5(2):100740. doi: 10.1016/j.xplc.2023.100740.). After the preliminary experiments on the cotton roots under saline-alkali stress, we found the compound saline-alkali stress could result in the more serious damages on the plant development. Besides, under less than 20 mins saline-alkali stress, cotton seedlings exhibited a certain degree of wilting, of which cotton tips could be utilized for further single-cell RNA-seq experiments. We have performed the single-cell RNA-seq on the cotton tips under saline-alkali stress, and those results will published in another article.

Reviewer 4 Report

Comments and Suggestions for Authors

Comprehensive General Comment

This manuscript investigates the role of GhERF2, an ethylene response factor, in the response of cotton (Gossypium hirsutum) to saline-alkali stress. By employing transcriptome analysis, gene ontology (GO), and KEGG enrichment analyses, the study identifies key genes and pathways involved in stress tolerance. The authors further validate the function of GhERF2 using molecular and physiological assays, including subcellular localization, gene silencing, and stress recovery experiments. The research provides insights into the molecular mechanisms of saline-alkali stress adaptation, with potential applications for improving stress tolerance in cotton breeding programs.

However, this manuscript needs to be improved as I will outline below.

Abstract

Lines 18–36

Comment: The abstract is concise and highlights the study's significance. However, it lacks quantitative data about the identified DEGs or GhERF2 gene’s impact.

Suggestion: Add specific data points (e.g., number of DEGs significantly related to saline-alkali stress or changes in physiological traits due to GhERF2 silencing).

Introduction

Lines 40–88

Comment: The introduction provides comprehensive background information but is verbose and contains redundant details about saline-alkali stress impacts.

Suggestion: Streamline the introduction by focusing more on the novelty of the research, such as the specific use of RNA-seq in identifying tissue-specific responses in cotton roots.

Materials and Methods

Lines 509–536

Comment: The hydroponic growth conditions and saline-alkali stress treatments are described, but details about replicates and controls are scattered.

Suggestion: Reorganize this section to clearly distinguish experimental groups, control setups, and replicates.

Lines 557–571

Comment: The description of RNA-seq data processing is thorough, but details about software versions and parameter settings are missing for some tools.

Suggestion: Specify the software versions and exact parameters used for data filtering and DEG identification to enhance reproducibility.

Results

Transcriptome Analysis

Lines 124–151

Comment: PCA and quality metrics are well-documented, but the inclusion of PCA plots in the supplementary material without detailed explanation in the main text reduces clarity.

Suggestion: Move key PCA insights to the main text and briefly explain their implications.

Functional Analysis

Lines 153–276

Comment: The presentation of GO and KEGG enrichment results is detailed but lacks biological interpretation of why certain pathways (e.g., plant hormone signal transduction) are significant under stress conditions. This point is very important for your manuscript.

Suggestion: Provide a deeper analysis of the enriched pathways' biological roles and their potential relevance to saline-alkali stress adaptation.

Validation of GhERF2

Lines 296–333

Comment: The subcellular localization of GhERF2 is well-illustrated but lacks explanation for why nuclear localization supports its function in stress response.

Suggestion: Discuss how nuclear localization ties into GhERF2's role as a transcription factor regulating stress-responsive genes.

Discussion

Lines 416–507

Comment: The discussion contextualizes findings within existing literature but misses opportunities to highlight the practical applications of GhERF2 in breeding programs.

Suggestion: Expand on how these findings can guide molecular breeding efforts for saline-alkali-resistant crops, potentially referencing successful applications of similar genes in other species. The value of your findings will increase significantly.

Figures and Tables

Comment: Figures effectively summarize the results, but some legends lack sufficient detail (e.g., Figure 9).

Suggestion: Include more descriptive figure legends to clarify experimental conditions, especially for complex visuals like heatmaps and Venn diagrams.

Author Response

Dear Editor and reviewers:

    On behalf of my co-authors, we thank you very much for your helpful efforts processing our manuscript entitled " Tissue-specific RNA-seq analysis of cotton roots response to compound saline-alkali stress and the functional validation of the key gene GhERF2" and providing us an opportunity to revise it. With regard to your and the reviewers’ positive and constructive comments on our manuscript, we think they are of high value and importance for improving our manuscript, as well as of critical guiding significance to our future researches.

After careful reviewing on your comments, we have made correspondent revisions in the manuscript, which we hope will meet the requirements of your journal. We have also sought for a professional help from a native specialist to improve the logicality and readability of our manuscript. A response to the reviewers’ comments is attached hereinafter for your review.

Thank you again for your kind help and efforts. If there is anything that need us do, please feel free to let us know.

Best regards, 

Pengtao Li

E-mail: lipengtao1056@126.com

Attachments:

Response to the comments of Reviewer 4

  1. Comment:The abstract is concise and highlights the study's significance. However, it lacks quantitative data about the identified DEGs orGhERF2 gene’s impact.

Response: Thank you very much for your valuable comments. We have made corresponding supplements to the abstract by adding the physiological trait changes caused by GhERF2 silencing. Please refer to lines 29-37: ...was identified to be associated with saline-alkali tolerance. Through virus-induced gene silencing (VIGS), the GhERF2-silenced plants exhibited a more severe wilting phenotype under combined salt-alkali stress, along with a significant reduction in leaf chlorophyll content and fresh weights of plants and roots. Additionally, these plants showed greater cellular damage and a lower ability to scavenge reactive oxygen species (ROS) when exposed to the stress. These findings suggest that the GhERF2 gene may play a positive regulatory role in cotton responses to salt-alkali stress.

  1. Comment:The introduction provides comprehensive background information but is verbose and contains redundant details about saline-alkali stress impacts.

Response: Thank you for your precious comments. We have streamlined the introduction by removing some redundant details about the effects of salt-alkali stress and providing a more detailed description of the specific use of RNA-seq in identifying tissue-specific responses in cotton roots. Please refer to lines 98-106: Notably, in a study focusing on cotton roots with different potassium ion affinities, a key regulatory target for K+ uptake under potassium deficiency were successfully identified using RNA-seq technology[35]. Previous studies also identified a key gene, GhMKK3, in cotton roots through RNA-seq and quantitative real-time PCR (qRT-PCR). This gene plays a role in drought stress response by regulating cotton stomatal behavior and root hair growth[36]. Gossypium hirsutum, a major cultivated cotton species, plays a vital role in cotton production and is a dominant species on saline lands [37].

  1. Comment:The hydroponic growth conditions and saline-alkali stress treatments are described, but details about replicates and controls are scattered.

Response: Thank you very much for your valuable feedback. We have reorganized this section to provide a clearer description of the experimental groups, control setups, and replicates. Please refer to lines 550-564: The hydroponic cotton seedlings obtained using the aforementioned germination method (two days after being transferred to hydroponic boxes) were used for transcriptome sequencing sample collection. The experimental group was treated with a combined salt-alkali solution for 20 minutes, while the control group was treated with distilled water for the same duration. After 20 minutes of salt-alkali stress treatment, cotton seedlings exhibited a certain degree of wilting. Therefore, samples were collected at this time point. Samples were then collected from root tips (with a standard length of 1 cm) and non-root tips of the cotton seedling roots. The experiment was performed in three biological replicates, each replicate consisting of 20 uniformly growing seedlings. All samples were rapidly frozen in liquid nitrogen and stored at -80℃ for subsequent transcriptome sequencing. The composition of the combined salt-alkali solution simulating the stress was designed based on the ionic composition of saline-alkali soil in Aral City, Tarim region, Xinjiang (see supplementary table), the soil ion composition was tested at Anyang Institute of Technology.

  1. 4.Comment:The description of RNA-seq data processing is thorough, but details about software versions and parameter settings are missing for some tools.

Response: Thank you very much for your great comments. We have supplemented the software versions and precise parameters used for data filtering and DEG identification. Please refer to lines 596-606:Expression levels of the genes were assessed by counting read numbers mapped to each gene using HTSeq v0.6.1[68], and FPKM was utilized for quantifying gene expression levels based on gene length and mapped read counts. Differential gene expression analysis was performed using DESeq2 between two different groups (and by the edgeR package in R version 4.4.0 for comparisons between two samples)[69], with differentially expressed genes/transcripts were set as the values of false discovery rate (FDR) less than 0.05 and absolute fold change equal to and more than 1.5. The functional enrichment of KEGG pathways and GO categories were analyzed using OmicShare (https://www.omicshare.com/tools/, accessed on 23 May 2024).

  1. 5.Comment:PCA and quality metrics are well-documented, but the inclusion of PCA plots in the supplementary material without detailed explanation in the main text reduces clarity.

Response: Thank you very much for your great comments. We have redrawn the PCA plot. Please refer to Supplementary Figure S1A.

  1. 6.Comment:The presentation of GO and KEGG enrichment results is detailed but lacks biological interpretation of why certain pathways (e.g., plant hormone signal transduction) are significant under stress conditions. This point is very important for your manuscript.

Response: Thank you very much for your valuable advice. We analyzed the biological roles of related enrichment pathways and their potential correlations with adaptation to salt-alkali stress. Please refer to lines 265-272: Plants may adjust the synthesis and distribution of hormones through plant hormone signal transduction pathways to cope with stress conditions. The MAPK signaling pathway is primarily involved in immune responses, environmental stress responses, and processes of growth and development. The cysteine and methionine metabolism pathways are believed to be closely associated with the synthesis of sulfur-containing metabolites and play a crucial role when plants are exposed to free radicals and other harmful compounds.

  1. 7.Comment:The subcellular localization of GhERF2 is well-illustrated but lacks explanation for why nuclear localization supports its function in stress response.

Response: Thank you very much for your careful and valuable reviewing. We have revised the corresponding parts of the manuscript according to your suggestions. Please refer to lines 314-319: Transcription factors are considered to play a crucial role in the transcriptional regulation processes within the cell nucleus, and nuclear localization is one of the key characteristics for determining whether a gene functions as a transcription factor or not. In plants, there are multiple transcription factor families, each playing distinct roles in stress responses.

  1. 8.Comment:The discussion contextualizes findings within existing literature but misses opportunities to highlight the practical applications of GhERF2 in breeding programs.

Response: Thank you very much for your worthy suggestion. We have referred to the applications of similar genes in other species, thereby further elucidating how these findings can guide the molecular breeding of salt-tolerant and alkaline-tolerant cotton varieties. Please refer to lines 518-522: Notably, certain members of the ERF subfamily in Krascheninnikovia arborescens and Xanthoceras sorbifolia have been shown to be closely associated with salt-alkali stress [65,66]. GsERF6 identified in soybean has been demonstrated to enhance tolerance of transgenic plants to salt-alkali stress[67].

  1. 9.Comment:Comment: Figures effectively summarize the results, but some legends lack sufficient detail (e.g., Figure 9).

Response: Thank you very much for your careful and valuable reviewing. We have revised the relevant sections to enhance the clarity of the figure legends in the manuscript.

Round 2

Reviewer 2 Report

Comments and Suggestions for Authors

Review 2 Plants

The authors have significantly improved the text of the manuscript, correcting the shortcomings contained in the first version. In their response, the authors explained the motives that prompted them to choose the experimental model used to study transcriptomic changes under salt stress and provided the manuscript with the missing Supplemental materials. However, they have not provided the raw data for transcriptome analysis which may be of great interest to readers. Despite this omission, the manuscript in its present form can be recommended for publication.

Author Response

1.Comment: The authors have significantly improved the text of the manuscript, correcting the shortcomings contained in the first version. In their response, the authors explained the motives that prompted them to choose the experimental model used to study transcriptomic changes under salt stress and provided the manuscript with the missing Supplemental materials. However, they have not provided the raw data for transcriptome analysis which may be of great interest to readers. Despite this omission, the manuscript in its present form can be recommended for publication.

Response: Thank you very much for your valuable comments. We have re-uploaded the raw data for transcriptome analysis, and the assigned accession of the submission is: CRA022812(https://bigd.big.ac.cn/gsa/browse/CRA022812).

Reviewer 3 Report

Comments and Suggestions for Authors

The manuscript has improved, but I still have some minor concerns.

  1. Figure 3: Please, change the caption to indicate “common DEGs”.
  2. Section 2.4: Given ~4,000 root specific DEGs (where I expect about 20% of all genes are DEGs), it is not surprising authors find DEGs in plan-associated pathways. Authors need to show only pathways that are highly enriched.  Also, Figures 4/5 need better annotation of samples (like what is GH_A02G0585).  In addition, genes like WRKY22/29 (in Figure 4) requires further explanation of how to merge data from multiple gene IDs (2 seems like DEGs while others are not).
  3. Section 2.6: Authors need to show the expressing changes of GhERF2 at least from the RNA-seq. Also, please include further explanation and references for GhERF2 from authors’ response.
  4. Figure 8: Axis label says “TRV:GhERF1”.

Author Response

  1. Comment:Figure 3:Please, change the caption to indicate “common DEGs”

Response: Thank you very much for your valuable comments. We have revised the caption of Figure 3 by replacing “DEGs” with “common DEGs”.

2.Comment: Given ~4,000 root specific DEGs (where I expect about 20% of all genes are DEGs), it is not surprising authors find DEGs in plan-associated pathways. Authors need to show only pathways that are highly enriched.  Also, Figures 4/5 need better annotation of samples (like what is GH_A02G0585).  In addition, genes like WRKY22/29 (in Figure 4) requires further explanation of how to merge data from multiple gene IDs (2 seems like DEGs while others are not).

Response: Thank you for your precious comments. We have selected the top three pathways with higher enrichment levels and constructed corresponding pathway diagrams, presenting the highly enriched pathways in the manuscript. To provide better annotation of the samples, we have included relevant content in lines 265-276 and lines 282-286 of the manuscript. Additionally, we have provided more detailed annotations for the legends of Figures 4/5.

3.Comment: Section 2.6: Authors need to show the expressing changes of GhERF2 at least from the RNA-seq. Also, please include further explanation and references for GhERF2 from authors’ response.

Response: Thank you very much for your valuable feedback.We have included information related to the expressing changes of GhERF2 from the RNA-seq in the manuscript.Please refer to lines 305-307.Herein is a further explanation regarding GhERF2:GhERF2 (ethylene-responsive transcription factor 2-like) belongs to the ERF subfamily [1,2]. Certain members of the ERF subfamily in Krascheninnikovia arborescens and Xanthoceras sorbifolia have been shown to be closely associated with salt-alkali stress [3,4]. Research using single-cell transcriptome sequencing has identified several candidate differentially expressed genes (DEGs) related to transcription factors and plant hormone responses to salt stress [5]. Among them, GaERF1A, an ethylene-responsive DEG related to salt stress, was identified in Gossypium arboreum and shares high homology with the key gene GhERF2 discussed in this manuscript. RNA-Seq analysis revealed that after 20 minutes of salt-alkali stress treatment on non-root tip and root tip samples, the expression level of GhERF2 was downregulated (Figure 6). Expression pattern analysis further showed that the gene's expression decreased within the first hour and then significantly increased over the next 24 hours (Figure 7C). Therefore, we suggest that the expression level of this gene is related to salt-alkali stress.Additionally, we further validated the role of the target gene in cotton's response to salt-alkali stress through experiments, including virus-induced gene silencing (VIGS).

4.Comment: Figure 8: Axis label says “TRV:GhERF1

Response: Thank you very much for your great commnets.We have made corresponding modifications to the caption and coordinate axes in Figure 8, replacing “TRV:GhERF1”with“TRV:GhERF2”.

References:

  1. ZafarMM,Rehman A,Razzaq A,Parvaiz  A,Mustafa G,Shari F, Mo H, Youlu Y,Shakee A, Ren M. Genome-Wide Characterization and Expression Analysis of Erf Gene Family in Cotton. BMC Plant Biol 2022, 22, 134.
  2. LuL,QanmberG,Li J, Pu M,Chen G, Li S,Liu L, Qin W,Ma S,Wang Y. Identification and Characterization of the ERF Subfamily B3 Group Revealed GhERF13.12 Improves Salt Tolerance in Upland Cotton. Front Plant Sci 2021, 12, 705883.
  3. 3. Wang J,Zhang Y,Yan X,Guo J. Physiological and Transcriptomic Analyses of Yellow Horn (Xanthoceras Sorbifolia) Provide Important Insights into Salt and Saline-Alkali Stress Tolerance. PLoS One 2020, 15, e0244365.
  4. 4. Zhang H,Wang Y,Ma B,Bu X,Dang Z,Wang Y. Transcriptional Profiling Analysis Providing Insights into the Harsh Environments Tolerance Mechanisms of Krascheninnikovia Arborescens. Int J Mol Sci 2024, 25, 11891.
  5. 5. LiP,LiuQ,Wei ; Xing C.; Xu Z.; Ding F.; Liu Y.; Lu Q.; Hu, N, Wang T. Transcriptional Landscape of Cotton Roots in Response to Salt Stress at Single-Cell Resolution. Plant Commun 2023, 5, 100740.
